# Sodium and Potassium Intakes and Cardiovascular Risk Profiles in Childhood Cancer Survivors: The SCCSS-Nutrition Study

**DOI:** 10.3390/nu12010057

**Published:** 2019-12-24

**Authors:** Fabiën N. Belle, Christina Schindera, Idris Guessous, Maja Beck Popovic, Marc Ansari, Claudia E. Kuehni, Murielle Bochud

**Affiliations:** 1Institute of Social and Preventive Medicine (ISPM), University of Bern, 3012 Bern, Switzerland; 2Center for Primary Care and Public Health (Unisanté), University of Lausanne, 1010 Lausanne, Switzerland; 3University Children’s Hospital Basel (UKBB), 4056 Basel, Switzerland; 4Pediatric Oncology, Children’s University Hospital of Bern, University of Bern, 3010 Bern, Switzerland; 5Division and Department of Primary Care Medicine, Geneva University Hospital HUG, 1205 Geneva, Switzerland; 6Pediatric Hematology-Oncology unit, Lausanne University Hospital CHUV, 1011 Lausanne, Switzerland; 7Pediatrics Onco-Hematology Unit, Geneva University Hospital HUG, 1205 Geneva, Switzerland; 8Cansearch Research laboratory, 1205, Geneva Medical School, 1205 Geneva, Switzerland

**Keywords:** sodium, potassium, nutrition, diet, urine spot, food frequency questionnaire, cardiovascular disease, childhood cancer survivors, Swiss Childhood Cancer Registry, Europe

## Abstract

Risk of cardiovascular disease (CVD), common in childhood cancer survivors (CCSs), may be affected by diet. We assessed sodium (Na) and potassium (K) intake, estimated from food frequency questionnaires (FFQs) and morning urine spots, and its associations with cardiovascular risk in CCSs. We stratified CCSs into three risk profiles based on (A) personal history (CVD, CVD risk factors, or CVD risk-free), (B) body mass index (obese, overweight, or normal/underweight), and (C) cardiotoxic treatment (anthracyclines and/or chest irradiation, or neither). We obtained an FFQ from 802 and sent a spot urine sample collection kit to 212, of which 111 (52%) returned. We estimated Na intake 2.9 g/day based on spot urine and 2.8 g/day based on FFQ; the estimated K intake was 1.6 g/day (spot urine) and 2.7 g/day (FFQ). CCSs with CVD risk factors had a slightly higher Na intake (3.3 g/day), than CCSs risk free (2.9 g/day) or with CVD (2.7 g/day, *p* = 0.017), and obese participants had higher Na intake (4.2 g/day) than normal/underweight CCSs (2.7 g/day, *p* < 0.001). Daily Na intake was above, and daily K intake below, the national recommended levels. Adult survivors of childhood cancer need dietary assistance to reduce Na and increase K intake.

## 1. Introduction

Cardiovascular disease (CVD) is the leading non-malignant cause of death in childhood cancer survivors (CCSs) [1,2,3]. CVD can be caused by cancer treatment and occurs prematurely in CCSs and increases with age without plateauing over time, leading to high morbidity and mortality [4]. Cardiotoxic cancer treatment includes chemotherapy with anthracyclines and chest irradiation [5,6]. Anthracyclines increase the risk of heart failure and chest irradiation is associated with heart failure, valvular disease, coronary artery disease, and atrial fibrillation [6,7]. In the US, CCSs are seven times as likely to die from a cardiac-related event than members of the general population [5]. Thirteen percent of CCSs exposed to cardiotoxic treatments suffered a life-threatening cardiovascular event in the 30 years after cancer treatment [8]. In the general population, hypertension, obesity, diabetes mellitus, smoking, and dyslipidemia are primary contributors to CVD [9,10,11]. These risk factors are also relevant for CCSs and represent important modifiable factors in the development and severity of CVD [12,13,14]. Efforts to reduce the obesity epidemic are an important pillar of cardiovascular prevention in the general population [9]. In CCSs, obesity management may substantially reduce the risk of premature cardiac risk: obese CCSs who received cardiotoxic treatment have a nine-times higher risk of developing coronary artery disease, higher than expected under an additive assumption [13].

Diets low in sodium (Na) and high in potassium (K) have been shown to reduce blood pressure and lower the risk of cardiovascular events and mortality in the general adult population [10,15,16,17]. The World Health Organization (WHO) recommends restricting Na intake (<2 g/day) and increasing K intake (>3.5 g/day) [16,17] but high sodium consumption is still common [15,18], particularly among obese persons [19,20], and may pose particular problems for CCSs, since their background CVD risk is higher. We know surprisingly little about the dietary intake of Na and K among CCSs, so we set out to assess Na and K intakes of CCSs based on FFQs and morning fasting urine spot samples. Urinary Na excretion, whether in spot urine or in 24 h collection, is a recognized biomarker of Na intake. In spot urine, the Na:creatinine (Cre) ratio needs to be used to account for urine concentration. Urinary Na:K ratio in spot or 24 h urine is a biomarker of Na and K intake that is strongly and positively associated with CVD risk. We determine if Na and K intakes are associated with CVD and modifiable cardiovascular risk factors.

## 2. Materials and Methods

### 2.1. Study Populations

#### The Swiss Childhood Cancer Survivor Study

The Swiss Childhood Cancer Survivor Study (SCCSS) is a population-based, long-term follow-up study of all childhood cancer patients registered in the Swiss Childhood Cancer Registry (SCCR) with leukemia, lymphoma, central nervous system tumors, malignant solid tumors, or Langerhans cell histiocytosis diagnosed in Switzerland. Participants were under the age of 21 years at the time of diagnosis, survived five years or more after the initial diagnosis of cancer, and were alive at the time of the study [21,22,23]. Ethical approval of the SCCSS and the SCCR was granted by the Ethics Committee of the Canton of Bern (KEK-BE: 166/2014); the SCCSS is registered at clinicaltrials.gov (NCT03297034).

CCSs were eligible to participate in the SCCSS-Nutrition study if they had childhood cancer diagnosed between 1976 and 2005, completed a baseline SCCSS questionnaire between 2007 and 2013 [21], and were 21 years of age or older at the time of follow-up survey in 2017. We sent a follow-up questionnaire that included an FFQ to all CCSs who were enrolled in SCCSS-Nutrition. We invited CCSs who returned the questionnaire and who lived in the French-speaking part of Switzerland to provide a urine spot sample (Figure 1). Detailed information on the SCCSS-Nutrition study design can be found in a prior publication [24].

### 2.2. Measurements

#### 2.2.1. Food Frequency Questionnaire

We assessed CCSs’ dietary intake, including information on portion sizes, with a self-administered, semiquantitative FFQ [26,27,28] originally developed and validated against 24-hour dietary recalls for the adult French-speaking Swiss population [25,26,29,30]. We extended the FFQ for CCSs with 15 additional food items, based on 24-hour dietary recall data from the Swiss National Nutrition Survey [31]. We investigated whether foods most frequently consumed in the German and Italian-speaking parts of Switzerland were also included in the FFQ and added 15 more food items in the FFQ that were reported more than 70 times (>2% of the total food items) during the 24-hour dietary recalls. The FFQ solicits information on consumption frequency and portion sizes for 112 fresh and prepared food items in 13 food groups (dietary supplements not included) during the four previous weeks. Consumption frequency ranges from “never during the last four weeks” to “two or more times per day”; portions can be equal to, smaller than or larger than a reference size. Reference portions were defined as common household measures representing the median portion size of a previous validation study performed with 24-hour dietary recalls [30]. The smaller and larger portion sizes represented the first and third quartiles of this distribution. When we used the FFQ before, we did not calculate daily Na and K intake [32], so we combined two sources to convert the food portions into macro- and micronutrients: the French Information Center on Food Quality (Centre d’Information sur la Qualité des Aliments, CIQUAL, Maisons-Alfort Cedex, France) and the Swiss Food Composition Database of the Federal Food Safety and Veterinary Office [25,33,34].

We calculated daily Na intake two ways: (1) by summing up the Na content of each FFQ food item, and (2) by using the equation developed specifically for this FFQ for males and females separately [25]:Males: (8.20 + 0.38 × Na (g/day) from FFQ)/2.54(1)
Females: (4.55 + 0.67 × Na (g/day) from FFQ)/2.54(2)
This equation is based on calibrations on total salt intake from 24-hour urine collections in a validation study that included 100 healthy people. We converted salt intake into Na intake, 1 g Na is equal to 2.54 g of salt (NaCl) [35].

#### 2.2.2. Sodium and Potassium Excretion Based on Morning Spot Urine

We asked CCSs living in the French-speaking part of Switzerland who returned the FFQ to collect a morning fasting urine spot sample and to send their sample by post to the pediatric hematology-oncology unit of Lausanne University Hospital, Lausanne, Switzerland. Upon arrival, technicians used validated routine laboratory procedures to measure levels of Na, K, and Cre. We used the International Cooperative Study on Salt, Other Factors, and Blood Pressure (INTERSALT) equation to estimate 24-hour urinary Na excretion from a morning fasting spot sample (Sp) as a marker for Na intake [36]. In this equation, urine Cre from the spot is needed to account for the urine concentration level of the spot.

We estimated 24-hour urinary Na excretion (g/day) for males and females:(3)$$\mathrm{Males}:\;23\times \frac{\left[25.46\;+\;0.46\;\times \;SpNa\;\left(\frac{mmol}{L}\right)–\;2.75\;\times \;SpCre\left(\frac{mmol}{L}\right)\;–\;0.13\;\times \;SpK\;\left(\frac{mmol}{L}\right)+\;4.10\;\times \;BMI\;\left(\frac{\mathrm{kg}}{m^{2}}\right)\;+\;0.26\;\times \;age\;\left(yrs\right)\right]}{1000}$$
(4)$$\mathrm{Females}:\;23\times \frac{\left[5.07\;+\;0.34\;\times \;SpNa\;\left(\frac{mmol}{L}\right)\;–\;2.16\;\times \;SpCre\;\left(\frac{mmol}{L}\right)–\;0.09\;\times \;SpK\left(\frac{mmol}{L}\right)\;+\;2.39\;\times \;BMI\;\left(\frac{\mathrm{kg}}{m^{2}}\right)+\;2.35\;\times \;age\;\left(yrs\right)-0.03\;\times \;age^{2}\right]}{1000}$$

We used the combined Pan American Health Organization (PAHO)/Chronic Kidney Disease Epidemiology Collaboration (CKD-EPI) equation to estimate 24-hour urinary K excretion (g/day) from a morning fasting spot sample as a marker for K intake [10,37]. PAHO, to estimate 24-hour urinary K excretion (g/day): $\frac{\left(\left(\frac{measured\;spot\;urine\;K\;}{measured\;spot\;urine\;Cre}\right)x\;estimated\;24-hour\;urinary\;Cre\;\right)x\;39.1}{1000}$, and CKD-EPI to estimate 24-hour urinary Cre excretion (mg/day): $879.89+12.51\;\times \;\mathrm{weight}\;\left(\mathrm{kg}\right)-6.19\;\times \;\mathrm{age}\;\left(\mathrm{yrs}\right)+\left(34.51,\;\mathrm{if}\;\mathrm{black}\right)-\left(379.42,\;\mathrm{if}\;\mathrm{female}\right)$.

#### 2.2.3. Sociodemographic, Lifestyle, and Clinical Characteristics

For all CCSs, we collected data on sex, age, country of birth, education, living situation, physical activity, smoking status, and body mass index (BMI) from the SCCR and the questionnaires. We calculated BMI by dividing weight in kilograms by height in meters squared (kg/m^2^) based on self-reported weight without clothes and height without shoes at the time of the FFQ survey. BMI was underweight (<18.5 kg/m^2^), normal (18.5–24.9 kg/m^2^), overweight (25–29.9 kg/m^2^), or obese (≥30 kg/m^2^) [38]. We also extracted additional clinical information from the SCCR including information on cancer diagnosis, age at diagnosis, and time since diagnosis. Diagnosis was classified according to the International Classification of Childhood Cancer, 3rd Edition (ICCC-3) [39]. Irradiation was classified as any irradiation or as cardiotoxic chest irradiation related to the original cancer diagnosis. Chest irradiation included mantlefield irradiation, irradiation of the thorax, mediastinum or thoracic spine, or total body irradiation. We collected a cumulative dosage of chest irradiation from medical records and categorized it based on the Children’s Oncology Group Long-Term Follow-up (COG-LTFU) guidelines for cardiology follow-up consultation. Chest irradiation was categorized as <30 Gray (Gy) or ≥30 Gy [40]. Other treatment exposures related to the original cancer diagnosis were divided into any chemotherapy, anthracycline-containing cardiotoxic chemotherapy, and hematopoietic stem cell transplantation (HSCT). We also retrieved records on relapse during follow-up.

#### 2.2.4. Cardiovascular Risk Profiles

We created three different cardiovascular risk profiles: (A) personal history of CVD and modifiable cardiovascular risk factors, and, specifically, (B) BMI at survey, and (C) cardiotoxic cancer treatment. We divided each risk profile into three severity levels. Personal history of CVD and modifiable cardiovascular risk factors (A) was split into: (1) “CVD” including heart attack, cardiomyopathy, angina pectoris, atrial fibrillation, arteriosclerosis, stroke, transient ischemic attack (TIA), and/or deep venous thrombosis; (2) “CVD risk factors” including hypertension (repeated high blood pressure measurements or antihypertensive medication treatment), obesity, diabetes mellitus treated with either tablets or insulin, current smoking, and/or high cholesterol defined as treatment with lipid-lowering medications, or (3) “CVD risk-free” if survivors did not report any of these conditions (Appendix A). Survivors could suffer from several CVD problems and CVD risk factors. BMI at survey (B) was categorized as (1) “Obese”, (2) “Overweight”, or (3) “Normal/underweight”. We categorized cardiotoxic cancer treatment (C) as (1) “Both anthracyclines and chest irradiation”, (2) “Either anthracyclines or chest irradiation”, or (3) “Neither anthracyclines nor chest irradiation”. Survivors were stratified into the three risk profiles (A, B, and C) based on their self-reports in the survey (A and B) or the information we extracted from the SCCR (C).

#### 2.2.5. Statistical Analyses

We included all CCSs who provided reliable dietary intake information [41] and were not pregnant or lactating during the FFQ survey (Appendix A). We compared the sociodemographic, lifestyle, and clinical characteristics of participants and nonparticipants to the SCCSS-Nutrition study. We also compared the dietary intake of participants to the FFQ and urine spot sample collection and between language regions within Switzerland. First, we assessed daily Na and K intake and determined if CCSs met the dietary DACH recommendations for Na and K intakes for Germany (D), Austria (A), and Switzerland (CH) [35]. To do this, we used both dietary assessment tools: the FFQ and the morning fasting spot urine. We used a Bland-Altman plot to determine the absolute agreement of Na and K intakes between the tools.

We then used analysis of covariance (ANCOVA) to assess differences in Na and K intakes (g/day) of CCSs across cardiovascular risk profiles A–C, adjusting for sex, age at survey (continuous), and ICCC-3 cancer diagnosis. We did not adjust initially for education level, smoking habits, physical activity, diet quality, or alcohol consumption because of our limited sample size and because these covariates may be affected by CVD, obesity, or cardiotoxic treatment (they are potential intermediates on the causal pathway). We tested if Na and K intake was approximately normally distributed for each category of the cardiovascular risk profile (profile A–C) by using the Shapiro-Wilk test of normality and skewness kurtosis test. To test for homoscedasticity, we used the Levene’s test for homogeneity of variances. We failed to reject a normal distribution for Na and K intake. There was also no issue with homoscedasticity. To compare differences between CVD risk groups, we calculated *p*-value from chi-square statistics (categorical, 2-sided test), analysis of variance (ANOVA, continuous, parametric), or the Kruskal-Wallis test (continuous, nonparametric). We determined the correlation between Na and K intakes (g/day) and self-reported BMI at FFQ survey (kg/m^2^).

All analyses were carried out with Stata (version 14, Stata Corporation, Austin, Texas). Statistical significance tests were two-sided with a significance level of 5%.

## 3. Results

Of 1749 eligible CCSs, we traced and contacted 1599, of whom 919 (57%) returned an FFQ (Appendix A). We excluded 11 survivors who were pregnant or breastfeeding, 35 who did not report their dietary intake, and 71 whose dietary intake data (<850 kcal or >4500 kcal per day) was unreliable [42]. We included 802 CCSs with FFQ data and sent 212 a spot urine sample collection kit; 111 morning fasting spot urine samples (52%) were returned to us.

Half of the CCSs were males; median age at FFQ survey was 35 years (interquartile range (IQR): 29–41 years): 21% were physically inactive, 16% smoked, and 9% were obese (Table 1). The mean alternative healthy eating index (AHEI) was 50.3 (95% CI: 49.1–51.5) for males and 56.0 (54.8–57.2) for females (maximum score = 110) (Table 1, Appendix A). Based on the self-reported personal history of CVD and cardiovascular risk factors (A), the groups differed in sociodemographic and lifestyle characteristics. CCSs who had CVD or CVD risk factors were more often males, were older, were less educated, lived alone, smoked, were obese, and consumed more alcohol than those who were CVD risk-free (p_all_ < 0.05, Table 1). The most common baseline cancers in CCSs had been leukemia, lymphoma, and central nervous system (CNS) tumors (Table 2). Median age at diagnosis was ten years (IQR: 4–14 years) and the median time from diagnosis to survey was 26 years (IQR: 20–32 years). More than a third of CCSs had received anthracyclines; 11% had chest irradiation. At baseline SCCSS questionnaire, 6% of the CCSs had CVD, 23% had CVD risk factors, and 71% were CVD risk-free. All CCSs who had indicated to have CVD at baseline still had CVD at the time of follow-up survey. Of those 187 CCSs who had CVD risk factors at baseline, 21 (11%) developed CVD, 118 (63%) stayed at risk, and 48 (26%) returned to CVD risk-free at follow-up. These 48 CCSs were no longer at CVD risk at follow-up survey since they had stopped smoking (90%), were no longer obese (6%), or had no longer hypertension (4%). At follow-up, 14% of the CCSs had CVD, 23% had CVD risk factors, and 62% were CVD risk-free.

Participants who filled in an FFQ were more often females, older, more likely to be born in Switzerland, tended to have a healthier lifestyle, and were younger at diagnosis than those who did not fill in an FFQ (Appendix A). Participants who returned a spot urine sample were older so time since their diagnosis was longer, but otherwise, they were similar to those who did not return a urine sample (Appendix A).

Na intake was higher than DACH recommends—when we used FFQ data it was 190% of the recommended level; it was 197% when we used spot urine samples (Table 3). Daily K intake was lower than DACH recommends—it was 68% when we used FFQ data and 40% when we used spot urine samples. Results for the subgroup taking part in the urine study (*n* = 111) were similar to the entire population (*n* = 802) and those who filled in the FFQ but did not provide a spot urine sample (*n* = 691; Appendix A). Na and K intake estimated by FFQ did not differ across language regions of Switzerland (Appendix A). Mean difference in daily Na intake, estimated by morning fasting spot urine and FFQ, was 0.2 g/day, and limits of agreement, of which 95% of the differences of spot urine samples compared to FFQ fall, were –1.2 and 1.5 g (Figure 1a). The mean difference in daily K intake, estimated by morning fasting spot urine and FFQ, was –1.0 g, and limits of agreement were –3.4 and 1.3 g (Figure 1b). Na intake was similar, whether based on FFQ or spot urine. K intake was lower when based on spot urine than when estimated by FFQ.

Na and K intakes did not differ between childhood cancer survivors across CVD risk profiles based on FFQ data (Table 4). Those with CVD risk factors had a slightly higher Na intake (3.3 g/day) calculated from morning fasting spot urine than those who were CVD risk-free (2.9 g/day) or those with CVD (2.7 g/day, *p* = 0.017, A); these results are in large part driven by obesity. Obese (4.2 g/day) or overweight (3.3 g/day) participants had a higher Na intake based on morning fasting spot urine than those who were normal/underweight (2.7 g/day, *p* < 0.001, B). Na intake estimated from spot urine was positively correlated with BMI (r = 0.57, Figure 2a). This was not seen for estimates based on FFQ (r = 0.16). We saw no relationship between K intake and BMI at the time of the survey, whether based on FFQ or spot urine (Figure 2b). We found no association of Na intake with cardiotoxic treatment groups (C), whether based on FFQ or spot urine data. The same was true for K. Additional adjustment for BMI at survey did not change our results for cardiotoxic treatment groups. Our results for Na and K intake across CVD risk profiles (A–C) did not change after additional adjustment for education level, smoking habits, physical activity, diet quality, or alcohol consumption.

## 4. Discussion

### 4.1. Principal Findings

Na and K intakes did not meet dietary recommendations among CCSs at a median of 26 years after participants’ cancer diagnosis. Na intake was almost double the recommended amount and K intake was about half the recommended amount. Na intake was particularly high in CCSs with CVD risk factors, and those who were overweight or obese. We saw no differences between cardiotoxic cancer treatment groups.

### 4.2. Strengths and Limitations

We used an FFQ that was not specifically designed to assess Na and K intakes, so we may have under- or overestimated dietary intake. To compensate for incorrect estimated daily dietary Na intake, we used a calibrated equation based on a validation study with 24-hour urine collection among Swiss adults [25]. After comparing the estimated Na intake assessed with FFQs using this equation and spot urine samples, we saw almost similar intake levels. To estimate Na intake from spot urine, we used the INTERSALT equation that includes urinary Cre to account for different levels of urine concentrations [36]. We took spot urine as a proxy of 24 h urine. The observed difference between K intake estimated from FFQ (2.7 g/day) and spot urine samples (1.6 g/day) could be due to over-reporting of fruit and vegetable consumption in the FFQ, the mean 196 ± 40 days between our two assessments [24], or diurnal variation of urinary K excretion. We asked CCSs for a single spot urine sample rather than multiple spot samples or 24-hour urine collections to minimize participation burden, although this is not considered the most accurate method to estimate daily Na intake. The collection of 24-hour urine is burdensome for CCSs, with a risk of incomplete collection and low response rates. Estimating Na and K excretions from morning fasting urine spots are a simple and cheap alternative, potential under- or over-collection is irrelevant and is considered to correlate well with 24-hour urine collections [43]. Yet, a single spot urine does not adequately capture intra-individual daily variations in Na and K intakes. Second, our data were based on a cross-sectional analysis, so that the captured dietary intakes represent current intakes and not past intakes. Dietary intake of CCSs with CVD or those with CVD risk factors could have changed over time due to, e.g., dietary recommendations from health care professionals, aging. We saw that survivors with CVD had a lower Na intake but a similar K intake, than those with CVD risk factors or who were CVD risk-free, suggesting that CCSs with CVD made salt reducing adaptations to their diet. However, only a few CCSs explicitly said that they followed a diet regimen because of an earlier CVD event. When looking at overall diet quality assessed with the AHEI, we saw similar scores between CCSs with CVD and those risk-free, whereas those with CVD risk factors scored slightly lower reflecting a poorer diet quality. Finally, all cardiac events as well as modifiable CVD risk factors were self-reported without medical record confirmation which could have led to incorrect reports. To improve validity, we went through the list of self-reported described medications to check if they were taken for antihypertensive, diabetes, and lipid-lowering medication treatment. CCSs were asked to report all their prescribed medication at the time of the FFQ survey.

Despite these limitations, the SCCSS-Nutrition study is strengthened by the large number of respondents who returned their FFQ and spot urine samples and our relatively long follow-up time after diagnosis. The SCCSS is strengthened by its national coverage, large sample size, and high response rate, which makes our results reasonably representative of this group nationwide. Furthermore, we had access to high-quality clinical information on cancer diagnosis and treatment extracted from the SCCR.

### 4.3. Na and K Intake and Cardiovascular disease (CVD): Results in Relation to Other Studies

CCSs [44,45,46,47,48,49] and the general population [15,25] often do not adhere to national or international dietary Na and K recommendations, so our finding is in line with those other studies. Our study showed a Na intake above dietary recommendations (males: 3.5 g/day, females: 2.2 g/day; 21–59 years), but a lower intake compared to the general Swiss population (males: 4.2 g/day, females: 3.2 g/day; 35–74 years) [25]. This difference may be due to the dissimilarity in age, as energy-adjusted salt intakes increases with age [25]. Excessive Na (143–155% the recommended level) and too low K (50–58% the recommended level) intakes were also observed in American CCSs [45,46]. Previous studies assessed Na and K intake based on FFQ data [45,47,49], 3-day food records [44,48], or 24-hour recalls [46]. These assessment methods have their limitations, e.g., underestimation of intake, recall bias [50], and may be subject to variability. Variability between methods may be due to the in- or exclusion of discretionary salt usage assessment like salt or salt-containing condiments added during cooking or the amount of food items questioned. Multiple 24-hour or spot urine collections could give us a better picture of real Na and K intake in this population. We found that CCSs who were overweight or obese had a higher estimated Na intake than normal/underweight CCSs, as in the general population [19]. The observed higher Na intake in obese CCSs might be associated with higher energy intake, fat metabolism, or insulin resistance [19,51].

There is strong evidence that reducing Na intake lowers blood pressure and hypertension risk [11]. Blood pressure response to Na intake is greater in children, adolescents, and adults with hypertension than in those without [15,52], so reducing Na intake in all CCSs, especially those who suffer from hypertension, could lower blood pressure and eventually reduce cardiovascular events and mortality. Because there were relatively few CCSs with hypertension in our sample (*n* = 21), we did not compare Na intake between CCSs with and those without hypertension; initially, we saw no significant differences in Na intake between these groups. Although optimal daily Na and K intake levels are still debated, moderate Na intake and high K intake appeared to be associated with the lowest risk of developing cardiovascular events and mortality in a study that included 103,570 participants from around the world [15]. Survivors might particularly benefit from dietary counseling, since they have an increased risk of developing CVD due to cardiotoxic cancer treatment [6,7]. Further research is needed to determine whether dietary intervention reduces cardiovascular events in CCSs at risk of developing CVD and if there are differences between those with and without hypertension.

### 4.4. Implications and Recommendations 

Pediatric oncology centers and follow-up clinics for CCSs in Switzerland follow the COG-LTFU guidelines, which advises a healthy lifestyle and medical management of modifiable cardiovascular risk factors in survivors at risk of CVD [53]. These guidelines include recommendations to consume a heart-healthy diet with avoidance of high salt intake [54]. Adherence to these guidelines may explain the lower daily Na intake observed in CCSs with CVD than in those who are CVD risk-free. Counseling survivors at risk of developing CVD to reduce their Na intake and increase their K intake may be an important intervention. Doctors should be supported by dieticians to improve other modifiable cardiovascular risk factors in CCSs such as overweight/obesity and dyslipidemia. Health care professionals and CCSs should be more aware about the increased CVD risk of CCSs to enhance CVD surveillance, early detection of asymptomatic CVD, routine screening and appropriate management of modifiable CVD risk factors, and to support diet and lifestyle modifications in an earlier stage to reduce CVD risk. Since CCSs have similar Na and K intakes as the general Swiss population, structural preventive measures are important as well. A first step could be population-based interventions promoting increased intake of potassium-containing foods, e.g., fruit, vegetables, nuts, discouraging the intake of salty foods and snacks, and reducing the salt content of processed and prepared foods by cooperating with the food industry.

## 5. Conclusions

We found poor dietary Na and K intakes in CCSs long after cancer diagnosis in Switzerland. High sodium intake was particular marked in those with CVD risk factors, or who were overweight or obese. Adult survivors of childhood cancer need dietary assistance to reduce Na and increase K intake. How this might specifically be done should be informed by further research into how CCSs may have acquired particular, post-treatment dietary tastes and habits.

## Figures and Tables

**Figure 1 nutrients-12-00057-f001:**
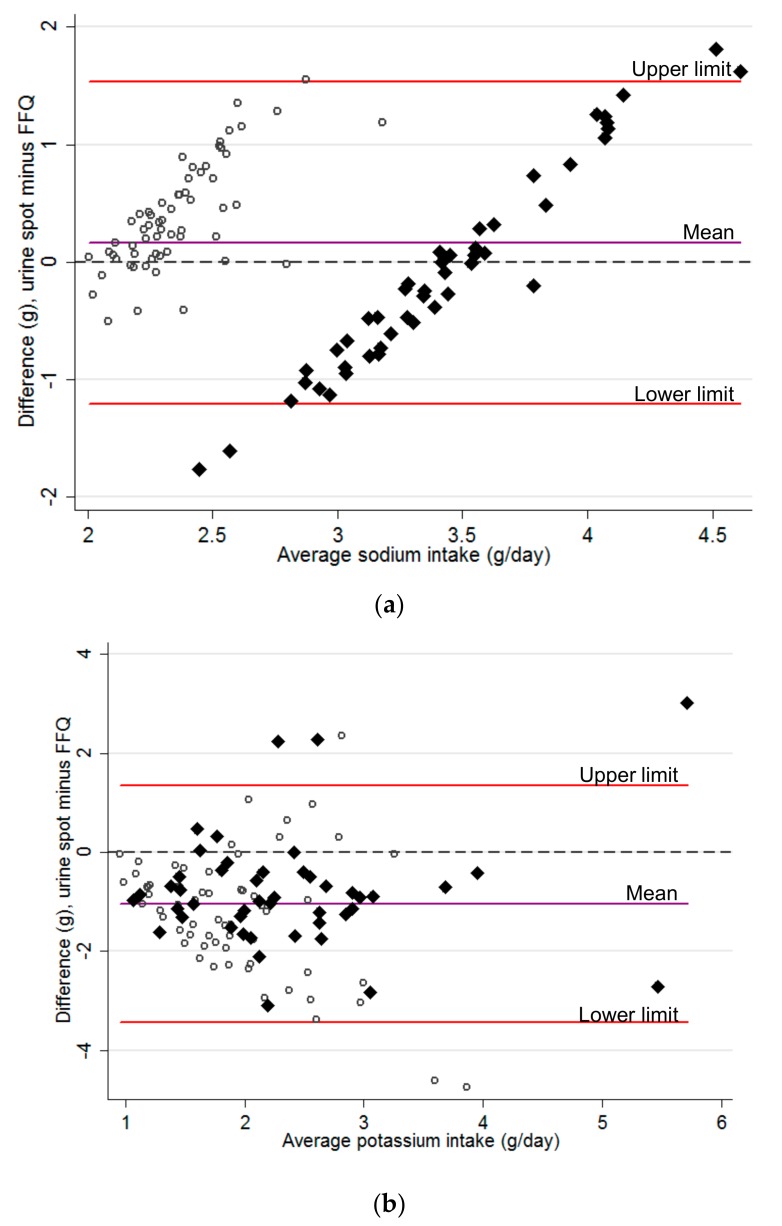
Bland–Altman plot of (**a**) average sodium and (**b**) potassium intake based on the intake calculated with a food frequency questionnaire ^1^ and a morning fasting spot urine in 111 childhood cancer survivors of the SCCSS-Nutrition study. FFQ—food frequency questionnaire; SCCSS—Swiss Childhood Cancer Survivor Study. Y = 0 is the line of perfect average agreement, the hollow circles (○) represent females and the black diamonds (♦) represent males. ^1^ Sodium (Na) intake based on food frequency questionnaire (FFQ) data was calculated with the equation: (8.20 + 0.38 × FFQ in males)/2.54 and (4.55 + 0.67 × FFQ in females)/2.54, with 1 g Na = 2.54 g salt (NaCl) [25]. Na intake is clustered by sex due to this equation. Intake based on FFQ data only gives a mean daily intake of 1.9 g ± 0.8.

**Figure 2 nutrients-12-00057-f002:**
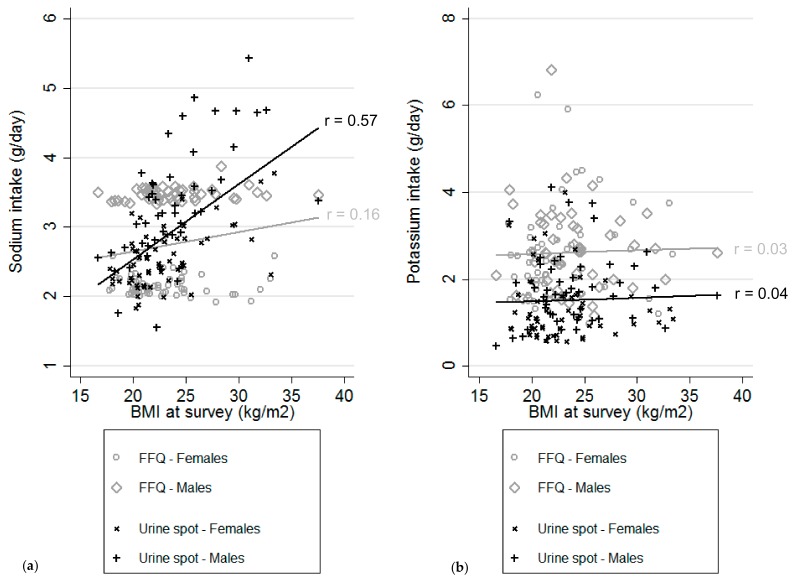
Cardiovascular risk profile B—correlation between BMI and sodium ^1^ (**a**) and potassium (**b**) measurements by food frequency questionnaire (FFQ) and morning fasting spot urine samples. BMI—body mass index; FFQ—food frequency questionnaire. ^1^ Sodium (Na) intake based on FFQ data was calculated with the equation: (8.20 + 0.38 × FFQ in males)/2.54 and (4.55 + 0.67 × FFQ in females)/2.54, with 1 g Na = 2.54 g salt (NaCl) see [25]. Intake based on FFQ data only gives a mean daily intake of 1.9 g ± 0.8.

**Table 1 nutrients-12-00057-t001:** Sociodemographic and lifestyle characteristics of adult childhood cancer survivors (CCSs) by cardiovascular risk profile (A).

	Total	Cardiovascular Risk Profile Based on (A) Personal History of CVD and Modifiable Risk Factors			
	CCSs *n* = 802	CVD ^1^ *n* = 114	CVD Risk Factors ^2^ *n* = 187	CVD Risk-Free *n* = 501	
	*n* (%)	*n* (%)	*n* (%)	*n* (%)	*p*-Value ^3^
**Sex**					
Females	401 (50)	64 (56)	74 (40)	263 (53)	0.004
Males	401 (50)	50 (44)	113 (60)	238 (48)	
**Age at survey**, median (IQR)	34.6 (28.8; 41.1)	38.0 (30.4; 45.2)	35.3 (29.5; 43.0)	34.1 (27.9; 39.7)	<0.001
≤30 y	248 (31)	28 (25)	54 (29)	166 (33)	<0.001
31–39 y	320 (40)	38 (33)	66 (35)	216 (43)	
≥40 y	234 (29)	48 (42)	67 (36)	119 (24)	
**Country of birth**					
Switzerland	763 (95)	111 (97)	173 (93)	479 (96)	0.119
Other	39 (5)	3 (3)	14 (7)	22 (4)	
**Language region within Switzerland**					
German speaking	570 (71)	82 (72)	123 (66)	365 (73)	0.480
French speaking	214 (27)	30 (26)	59 (32)	125 (25)	
Italian speaking	18 (2)	2 (2)	5 (3)	11 (2)	
**Education (highest degree)**					
Lower than university	532 (66)	77 (68)	142 (76)	313 (62)	0.004
University	270 (34)	37 (32)	45 (24)	188 (38)	
**Living situation**					
Alone	164 (20)	24 (21)	52 (28)	88 (18)	0.012
With others	638 (80)	90 (79)	135 (72)	413 (82)	
**Physical activity** ^4^					
Inactive	165 (21)	26 (23)	39 (21)	100 (20)	0.790
Active	637 (79)	88 (77)	148 (79)	401 (80)	
**Smoking status**					
Current	128 (16)	15 (13)	113 (60)	-	<0.001
Former	132 (16)	24 (21)	11 (6)	97 (19)	
Never	542 (68)	75 (66)	63 (34)	404 (81)	
**BMI at survey**					
Obese, ≥30 kg/m^2^	75 (9)	13 (11)	62 (33)	-	<0.001
Overweight, ≥25 to <30 kg/m^2^	177 (22)	26 (23)	43 (23)	108 (22)	
Normal, ≥18.5 to <25 kg/m^2^	511 (64)	72 (63)	76 (41)	363 (72)	
Underweight, <18.5 kg/m^2^	39 (5)	3 (3)	6 (3)	30 (6)	
**AHEI**^5^, mean ± SD	53.1 ± 12.7	53.7 ± 10.8	51.3 ± 12.4	53.7 ± 13.1	0.089
**Alcohol**^6^, mean drinks/day ± SD	0.43 ± 0.61	0.36 ± 0.50	0.58 ± 0.75	0.39 ± 0.56	<0.001

AHEI—alternative healthy eating index; BMI—body mass index; CCSs—childhood cancer survivors; IQR—interquartile range; SD—standard deviation. ^1^ This includes 69 CCSs with atrial fibrillation, 21 with deep venous thrombosis, 20 with cardiomyopathy, 10 with stroke/ transient ischemic attack (TIA), 8 with angina pectoris, 5 with a heart attack, and 5 with arteriosclerosis. ^2^ This includes 128 CCSs who are current smokers, 75 with obesity, 21 with repeated high blood pressure, 13 with high cholesterol, and 8 with diabetes mellitus treated with either tablets or insulin. ^3^
*p*-value calculated from chi-square statistics (categorical, 2-sided test), analysis of variance (ANOVA, continuous, parametric) or the Kruskal-Wallis test (continuous, nonparametric) to compare differences between CVD risk groups. ^4^ Active: ≥150 min of moderate intense or 75 min of vigorous intense or a combination of moderate and vigorous intense physical activity per week. ^5^ Adapted from Chiuve et al. J Nutr 2012 142(6):1009-18. Adjusted for age at survey and cancer diagnosis based on the International Childhood Cancer Classification, 3rd edition. For more information see [32]. ^6^ One serving was 113.4 g of wine, 340.2 g of beer or 42.5 g of liquor.

**Table 2 nutrients-12-00057-t002:** Clinical characteristics of adult childhood cancer survivors (CCSs) by cardiovascular risk profile (A).

	Total	Cardiovascular Risk Profile Based on (A) Personal History of CVD and Modifiable Risk Factors			
	CCSs *n* = 802	CVD ^2^ *n* = 114	CVD Risk Factors ^3^ *n* = 187	CVD Risk-Free *n* = 501	
	*n* (%)	*n* (%)	*n* (%)	*n* (%)	*p*-value ^4^
**ICCC3 diagnosis**					
I: Leukemia	246 (31)	35 (31)	55 (29)	156 (31)	0.189
II: Lymphoma	173 (22)	25 (22)	42 (22)	106 (21)	
III: CNS tumor	81 (10)	7 (6)	28 (15)	46 (9)	
IV: Neuroblastoma	28 (3)	5 (4)	6 (3)	17 (3)	
V: Retinoblastoma	12 (2)	-	4 (2)	8 (2)	
VI: Renal tumor	52 (6)	7 (6)	8 (4)	37 (7)	
VII: Hepatic tumor	6 (<1)	3 (3)	1 (<1)	2 (<1)	
VIII: Bone tumor	50 (6)	7 (6)	10 (5)	33 (7)	
IX: Soft tissue sarcoma	66 (8)	13 (11)	8 (4)	45 (9)	
X: Germ cell tumor	43 (5)	8 (7)	10 (5)	25 (5)	
XI & XII: Other tumors	26 (3)	3 (3)	9 (5)	14 (3)	
Langerhans cell histiocytosis	19 (2)	1 (<1)	6 (3)	12 (2)	
**Age at diagnosis**^1^ years	9.7 (3.9; 13.9)	10.8 (5.1; 15.5)	10.3 (4.0; 13.8)	8.8 (3.8; 13.7)	0.232
**Time since diagnosis**^1^ years	26.1 (20.2; 31.7)	28.5 (21.3; 34.7)	26.5 (20.8; 32.3)	25.3 (19.8; 30.7)	0.002
**Irradiation, any**	292 (36)	46 (40)	64 (34)	182 (36)	0.562
Chest irradiation	87 (11)	14 (12)	14 (7)	59 (12)	0.238
<30 Gy	31 (4)	6 (5)	4 (2)	21 (4)	0.486
≥30 Gy	56 (7)	8 (7)	10 (5)	38 (8)	
**Any chemotherapy**	642 (80)	98 (86)	142 (76)	402 (80)	0.106
Anthracyclines	298 (37)	46 (40)	61 (33)	191 (38)	0.309
**HSCT**	30 (4)	6 (5)	6 (3)	18 (4)	0.634
**History of relapse**	99 (12)	16 (14)	23 (12)	60 (12)	0.833

CCSs—childhood cancer survivor; CNS—central nervous system; CVD—cardiovascular disease; Gy—gray; HSCT—hematopoietic stem cell transplantation; ICCC-3—International Childhood Cancer Classification, 3rd edition, IQR—interquartile rang; NA—not applicable ^1^ Values are medians (IQRs). ^2^ This includes 69 CCSs with atrial fibrillation, 21 with deep venous thrombosis, 20 with cardiomyopathy, 10 with stroke/ transient ischemic attack (TIA), 8 with angina pectoris, 5 with a heart attack, and 5 with arteriosclerosis. ^3^ This includes 128 CCSs who are a current smoker, 75 with obesity, 21 with repeated high blood pressure, 13 with high cholesterol, and 8 with diabetes mellitus treated with either tablets or insulin. ^4^
*p*-value calculated from chi-square statistics (categorical, 2-sided test) or the Kruskal-Wallis test (continuous, nonparametric) to compare differences between CVD risk groups.

**Table 3 nutrients-12-00057-t003:** Sodium (Na) and potassium (K) intake of adult childhood cancer survivors (CCSs).

	DACH Recommendations ^1^	CCSs			
		Based on FFQ *n* = 802		Based on Morning Fasting Spot Urine *n* = 111	
		Mean ± SD	% DACH ^3^	Mean ± SD	% DACH ^3^
Na, g	1.5	2.8 ± 0.7 ^2^	190	2.9 ± 0.8 ^4^	197
K, g	4.0	2.7 ± 1.1	68	1.6 ±1.0 ^5^	40
Na:K ratio ^6^	*-*	1.2 ± 0.5	*-*	2.3 ± 1.0	*-*
Na:Cre ratio	*-*	*-*		2.3 ± 0.4 ^7^	*-*
K:Cre ratio	*-*	*-*		1.2 ± 0.7 ^7^	*-*

CCS—childhood cancer survivors, Cre—creatinine; DACH—dietary recommendations for Germany (D), Austria (A) and Switzerland (CH); Na—sodium; K—potassium. ^1^ DACH recommendations for the general population age 20–50 years, excluding pregnant and lactating females, 2015. Recommendations for Na and K are similar for males and females [35]. ^2^ Na intake based on FFQ data was calculated with the equation: (8.20 + 0.38 × FFQ in males)/2.54 and (4.55 + 0.67 × FFQ in females)/2.54, with 1 g Na = 2.54 g salt (NaCl) see [25]. Intake based on FFQ data only gives a mean daily intake of 1.9 g ± 0.8. ^3^ Percentage of mean intake in relation to the DACH recommended intake level × 100. Recommended intake is estimated on the basis of the age–sex groups of the DACH guidelines, weighted by the age and sex distribution of the study population [35]. ^4^ Calculated based on the International Cooperative Study on Salt, Other Factors, and Blood Pressure (INTERSALT) equation [36]. ^5^ Calculated based on the combined Pan American Health Organization (PAHO)/Chronic Kidney Disease Epidemiology Collaboration (CKD-EPI) equation [10,37]. ^6^ Na:K ratio, as an indicator for CVD risk, was calculated by dividing estimated Na intake (g/day) by estimated K intake (g/day). A high Na:K ratio is associated with an increased risk of CVD. ^7^ Na/K: Cre ratio was calculated using the INTERSALT, the combined PAHO/CKD-EPI, and CKD-EPI equation [10,36,37].

**Table 4 nutrients-12-00057-t004:** Mean sodium (Na) and potassium (K) intake (g/day) in childhood cancer survivors by cardiovascular risk profiles (A–C), retrieved from ANCOVA ^1.^

	Based on FFQ *n* = 802					Based on Morning Fasting Spot Urine *n* = 111				
	*n* (%)	Na ^2^ (95% CI)	*p*	K (95% CI)	*p*	*n* (%)	Na (95% CI)	*p*	K (95% CI)	*p*
Cardiovascular risk profile										
**(A) Personal history of CVD and risk** **factors**										
CVD	114 (14)	2.9 (2.8, 2.9)	0.538	2.7 (2.5, 2.9)	0.058	16 (14)	2.7 (2.3, 3.0)	0.017	1.3 (0.8, 1.8)	0.490
CVD risk factors	187 (23)	2.8 (2.8, 2.9)		2.6 (2.4, 2.7)		28 (25)	3.3 (3.0, 3.6)		1.7 (1.3, 2.1)	
CVD risk-free	501 (62)	2.8 (2.8, 2.9)		2.8 (2.7, 2.9)		67 (60)	2.9 (2.7, 3.0)		1.6 (1.4, 1.8)	
**(B) Self-reported BMI at survey**										
Obese, ≥30 kg/m^2^	75 (9)	2.9 (2.8, 2.9)	0.355	2.6 (2.3, 2.8)	0.296	10 (9)	4.2 (3.8, 4.6)	<0.001	2.1 (1.5, 2.7)	0.272
Overweight, ≥25 to <30 kg/m^2^	177 (22)	2.9 (2.8, 2.9)		2.8 (2.6, 2.9)		15 (14)	3.3 (3.0, 3.6)		1.6 (1.1, 2.1)	
Normal/underweight, <25 kg/m^2^	550 (69)	2.8 (2.8, 2.9)		2.8 (2.7, 2.9)		86 (77)	2.7 (2.6, 2.9)		1.5 (1.3, 1.7)	
**(C) Cardiotoxic treatment**										
Both anthracyclines and chest irradiation	51 (6)	2.9 (2.8, 2.9)	0.481	3.1 (2.7, 3.4)	0.082	9 (8)	3.3 (2.8, 3.9)	0.349	1.8 (1.1, 2.5)	0.518
Either anthracyclines or chest irradiation	283 (35)	2.8 (2.8, 2.9)		2.8 (2.6, 2.9)		36 (32)	3.0 (2.7, 3.2)		1.7 (1.4, 2.1)	
Neither anthracyclines nor chest irradiation	468 (58)	2.8 (2.8, 2.9)		2.7 (2.6, 2.8)		66 (59)	2.9 (2.7, 3.1)		1.5 (1.2, 1.7)	

ANCOVA—analysis of covariance; BMI—body mass index; CVD—cardiovascular disease; FFQ—food frequency questionnaire; K—potassium; Na—sodium. ^1^ Adjusted for sex, age at survey, and ICCC-3 cancer diagnosis. ^2^ Na intake based on FFQ data was calculated with the equation: (8.20 + 0.38 × FFQ in males)/2.54 and (4.55 + 0.67 × FFQ in females)/2.54, with 1 g Na = 2.54 g salt (NaCl) see [25]. Intake based on FFQ data only gives a mean daily intake of 1.9 g ± 0.8.

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
