# Peer review of "Sodium and Potassium Intakes and Cardiovascular Risk Profiles in Childhood Cancer Survivors: The SCCSS-Nutrition Study"

_nutrients, 2019, doi:10.3390/nu12010057_

Round 1

Reviewer 1 Report

Thank you for allowing me to review the manuscript examining Na and K levels in childhood cancer survivors. Below you will find my comments and suggestions.

1. For past cancer treatment, was that treatment for the childhood cancer diagnosis only or does that include any treatment up until FFQ? Childhood cancer survivors are at risk for second malignancies and may receive additional cancer treatment in survivorship.

2. There is very limited details given regarding the self-report of CVD and CVD risk factors. For example, it is not clear or commented on how long ago participants may have been diagnosed with CVD or CVD risk factors. How many of these CVDs were present when survivors were enrolled at baseline, before baseline assessment? Please add details regarding time since CVD diagnosis if possible.

3. The first mention of creatinine is in the methods section when discussing the urine spot test. The relevance of creatinine was not mentioned in the introduction, methods, or discussion. Please expand on why creatinine was included.

4. The ratio between Na:K is a risk factor for CVD. Please add brief details as to why Na:K is measured as an indicator.

5. It was not clear if there were there participants who had more than one CVD? Please clarify the burden of disease.

6. In the "Sociodemographic, Lifestyle, and Clinical Characteristics" methods section it states BMI was calculated based on height and weight taken at time of FFQ. In the following section for "Cardiovascular Risk Profiles" you state that BMI was self-reported at survey. And in the "Statistical Analysis" section it states that BMI was self-reported. It is not clear why there were two measurements of BMI?

7. In line 101, it states "When we used the FFQ before, we did not include daily Na and K intake". Please clarify.  

Reviewer 2 Report

The paper is very interesting and on an important topic regarding long term health of childhood cancer survivors. The paper is well written and I only recommend some minor revisions. The introduction is well written and sharpy focused on the topic. Methods: The study is well planned and conducted, there do not seem to be any bigger flaw in the design. Thus, the authors should give some information on the post hoc power / detectable effect size. Regarding the statistical analysis, it stays unclear why the authors used categorized variables instead of the continuous. Particularly, the binarization of radiation exposures seem to be very artificial. The author shoud give a statment for using these categories or, more preferable, maybe use an ordinal regression model for analysis, with countinuous covariables. Within the methods section, the authors only describe the use of ANCOVA. According to their tables, they also used Chi² and Kruskal-Wallis. The authors should therefore add information on all used statistical method and how desicion for a test (especially choosing ANCOVA or Kruskal-Wallis) was done. If the authors tested for distribution and homoscedastic, please explain!
